# Internet-of-Things-Assisted Smart System 4.0 Framework Using Simulated Routing Procedures

**Jinglei Su [1], Xue Chu [2,*], Seifedine Kadry [3,*] and Rajkumar S [4]**

[1]   Business School, Jiangsu Normal University, Xuzhou 221116, China; sujl666@jsnu.edu.cn
[2]   State-owned Assets Management Department, Qilu Normal University, Jinan 250200, China
[3]   Department of Mathematics and Computer Science, Beirut Arab University, Beirut 115020, Lebanon
[4]   School of Computer Science and Engineering, Vellore Institute of Technology, Vellore 632014, India; rajkumarsrajkumar@gmail.com
*   Correspondence: chuxue2019@qlnu.edu.cn (X.C.); s.kadry@bau.edu.lb (S.K.)

**Abstract:** The environment and energy are two important issues in the current century. The development of modern society is closely linked to energy and the environment. Internet of Things (IoT) and Wireless Sensor Networks (WSNs) have recently been developed substantially to contribute to the fourth transformation of the power grid, namely the smart grid. WSNs have the potential to improve power grid reliability via cable replacements, fault-tolerance features, large-scale protection, versatility to deploy, and cost savings in the smart grid environment. Moreover, because of equipment noise, dust heat, electromagnetic interference, multipath effects, and fading, current WSNs are making it very difficult to provide effective communication for the smart grid (SG) environment, in which WSN work is more difficult. For the smart system 4.0 framework, a highly reliable communication network based on the WSN is critically important for the successful operation of electricity grids in the next decade. To solve the above problem, a Robust Bio-Dynamic Stimulated Routing Procedure (RDSRP) has been proposed based on the real-time behavior of a new Hybrid Bird Optimizer (HBO) model. The presented innovative research and development is a small yet important aspect of continuous critical activities that address one of our society's major challenges and that reverse the dangerous trends of environmental destruction. This study explores some of the most recent advances in this area, including energy efficiency and energy harvesting, which are expected to have a significant impact on green topics under smart systems in the Internet of things. The experimental results show that the proposed distributed system suggestively enhances network efficiency and reduces the transmission of excess packets for wireless sensor network-based smart grid applications.

**Keywords:** Internet of Things (IoT); Wireless Sensor Networks (WSNs); smart grid; dynamic stimulated routing procedure

## 1. Introduction

A fourth step of the industrial revolution called the smart system 4.0 framework has been made possible by recent advances in Communications and Information Technology (CIT) [1]. The key concepts of term 4.0, as the high-tech plan for 2020 in Germany [2], were explored for the first time in 2011. Intelligent application and regulation of local and global physical manufacturing processes using advanced CITs form a theoretical basis for the smart system 4.0 framework [3].CITs, therefore, play a leading role in increasing overall versatility to increase productivity in terms of manufacturing capital [4]. In the smart system 4.0 framework, customers would be able to select a range of goods at

desired values at constant prices, at a high level of liberty, and at anywhere around the world [5]. It is certainly possible that the number of retail buyers and sellers and those in the lifeworld will rise [6].

To do this, the Internet of things has the main purpose, through the internet of services, of linking different kinds of cyber-physical systems [7], of creating, and of interacting and coordinating closely with people [8]. To collect and exchange data in real-time data on the Internet, intelligent analysis and control of logic from all over the world are facilitated [9].

As inferred from Figure 1, Domain Access software is a software application which provides full life cycle access for remote management to the domain Operating system (OS) and the Base Management Card (BMC) in the data center. Power cycling helps users to control servers remotely while they are disabled and to turn up/down or to restart servers while they are operating the system OS based on web service systems. Cloud access requires all on- and off-band domain control based on the protocol control panel. To address hardware problems preventing the server from booting normally, BIOS (base input/output system) or BMC are accessed directly from the device program. This data and expertise are used efficiently and efficiently and shift the relationships between market makers, consumers, and stakeholders [10]. In contrast, the cost of services and cloud storage [11], such as internet stores, big data analytics, and web portals, decreases or increases, which is shown in Figure 1. In addition, using Internet of Things (IoT) in 4.0 would increase production efficiency and would thus lead to substantial financial benefits in making smart devices [12–16]. The improved flexibility of the smart grid allows greater penetration of highly variable sources of renewable energy, such as solar power, automated energy distribution, residential energy utilization, renewable resources, and wind power, even without energy storage. The current network technology is not intended for a vast number of dispersed feed-up points, and typically, the system at the receiving stage cannot support it, even though any feed-in is allowed at the local (distributive) stage. Fast variations, for instance, due to cloudy or rainy weather, are a major challenge for power engineers who need to ensure stable electricity by changing the outputs of more controllable generators, such as gas turbines and hydraulic generators. For this reason, an intelligent grid technology is a necessary prerequisite for very large amounts of renewable energy in the grid.

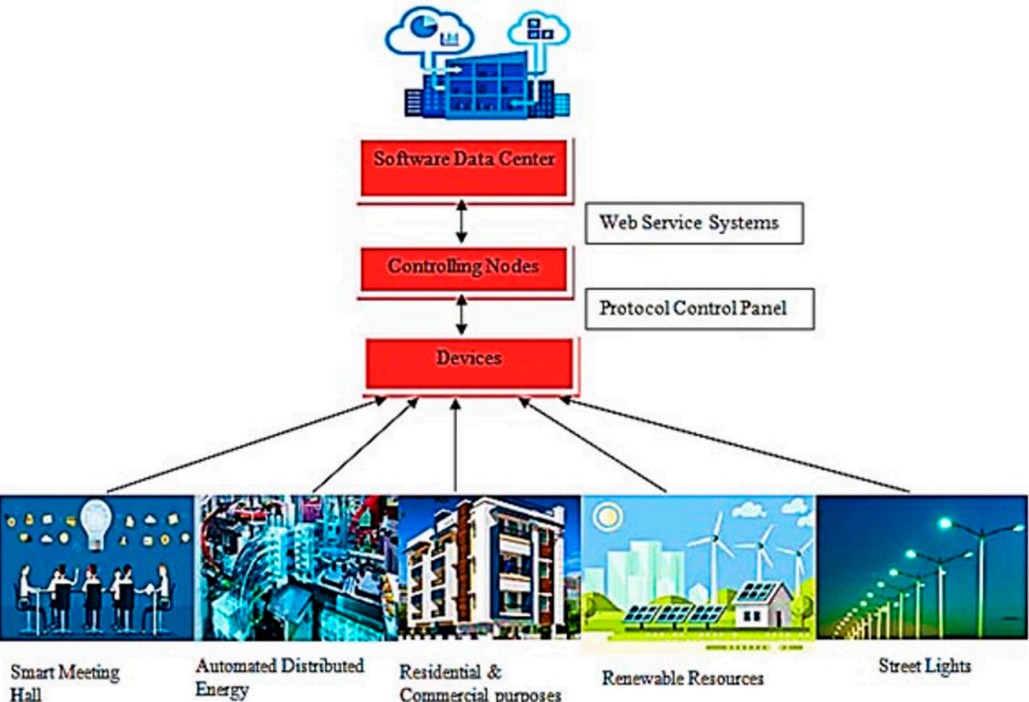

**Figure 1.** Control network for smart grid energy system.

The key objectives of the two CITs on the communication side are to make it possible for 4.0 to automate information exchange between different production systems [17] and to ensure highly stable networks in intelligent factories. Due to the security and complex service quality (QoS), specifications of different factories and the process of implementation are difficult [18–24]. Wireless networking is a complementary approach, as opposed to a wired network, to enable factory component control and management skills [25]. The contributions in the paper are explained as follows:

- This paper proposes the Robust Bio-Dynamic Stimulated Routing Procedure (RDSRP) based on the real-time behavior of a new Hybrid Bird Optimizer (HBO) model. The proposed bio-inspired model can identify the nearest heavily used grids in the sense of optimization-based machining, butterfly operation, and genetic changes in the maturing process.
- In addition, double sinks are used to spread the data traffic burden dramatically and to reduce memory overload and problems in the network sensor nodes using packet transmission gully along a narrow roadway in each subregion.

The suggested routing procedure, however, avoids unnecessary track failure and uses a smart self-learning process to deal with problems of the network tracking failure on time. For sparse and densely spread SG implementations, the proposed routing procedure is ideal. The remainder of the paper is organized as follows: Section 2 briefly studies the related works that have been proposed in the past and their pros and cons. Section 3 discusses the proposed methodology with an explanation of device and communication security. In addition, a Robust Bio-Dynamic Stimulated Routing Procedure (RDSRP) has been proposed based on the real-time behavior of a new Hybrid Bird Optimizer (HBO) model. In Section 4, the performance analysis of the proposed method is discussed with the simulation settings and appropriate comparisons, followed by a conclusion in Section 5.

## 2. Literature Survey

A channel allocation Clustering-Based Approach (CBA) has been proposed [26] for users that take into account practical restrictions in the SG context. Next, the basic communication model of SG networks is based on CR (Cognitive Radio). Depending on the distance from the smart grid, they split the service area into code groups called district network clusters. A multiple NP-hard (Nondeterministic Polynomial time) CA (Cognitive Approach) problem is developed using a strategy for avoiding disputes between the fairness distribution and priorities of two realistic scenarios. They suggest our CA algorithm, based on cat swarm optimization, to remove the extreme integer limitations of the problem. The results show that our proposed CBA algorithm works correctly, both in fair and priority circumstances.

Fish Bone Routing (FBR), proposed in [27], is a robust loop-free data routing system to avoid high transport costs, overtime, and unnecessary data transmission from the multi-hop network. FBR facilitates reliable loop-free data paths and prevents high costs for transmission, unreasonable delays from end to end, and the wasteful transfer of multi-hop data from the source to the network destination [14]. This reduces the likelihood of a data packet failure dramatically and preserves reliable networking properties between sensor nodes to balance loads and to extend the lifespan of wireless sensor grids in harsh, intelligent grid environments.

The system allows power utilities to track and control energy generation, transmission, and distribution processes more efficiently, flexibly, securely, reliably, distributed, safely, and cost-effectively through the use of state-of-the-art Intelligent Information Processing (IIPs) [28]. SG is faced with many obstacles in the sense of Smart System 4.0 (SS 4.0), although it provides many opportunities. In this regard, this paper provides a detailed review in the sense of the smart system 4.0 framework on essential smart grid components with international standards and information technology. This study provides a report on the benefits, disadvantages, and specifications of various smart grid applications.

In this paper, the IWSN (Industrial Wireless Sensor Network) [29] paradigms are a guiding tool for automation of the industrial grids today and even for the fourth step in the industrial

revolution, called Smart Net Industry (SGI) 4.0. Nonetheless, the dynamic smart grid (SG), due to noise, electromagnetically interferences, multi-channel effects, and decompression in SG settings, poses major challenges for secure interaction in SG applications. The service quality (QoS) and network life of the multi-hop WSNs are therefore declining based on interactions with SG applications [30]. For the smart system 4.0 framework, it is important to have a very secure communication infrastructure at WSN to connect wirelessly and to incorporate power system components to make the operation of next-generation grids more efficient, more competitive, and intelligent.

Cognitive Radio Sensor Networks (CRSN) [31] is introduced as a sound, reliable, and effective communication system capable of addressing both current and future smart grid energy management requirements. This paper suggests CRSN algorithms for channel allocation to implement these applications. The frame established considerably reduces the likelihood of packet loss and preserves the high communication performance of sensor nodes in harsh intelligent grids. In terms of product availability, time, and energy consumption, the proposed solution output has been tested and shown to successfully meet the QoS criteria of most of the presented SG applications [31].

From the above, it is proven that the existing methods are lacking in spreading the data and in increasing the traffic burden dramatically. Memory is increased due to several packet transmission data.

## 3. Robust Bio-Dynamic Stimulated Routing Procedure (RDSRP) Based on the Real-Time Behavior of a New Hybrid Bird Optimizer (HBO) Model

For people of the world, engineering is always useful, as shown in Figure 2. Engineering helps people live with healthy living standards in all forms of fields associated with humans. People in the past realized that slow working strategies are expected to be used. Such tools are produced only by hand. It takes a lot of time to do work or to work with such living styles.

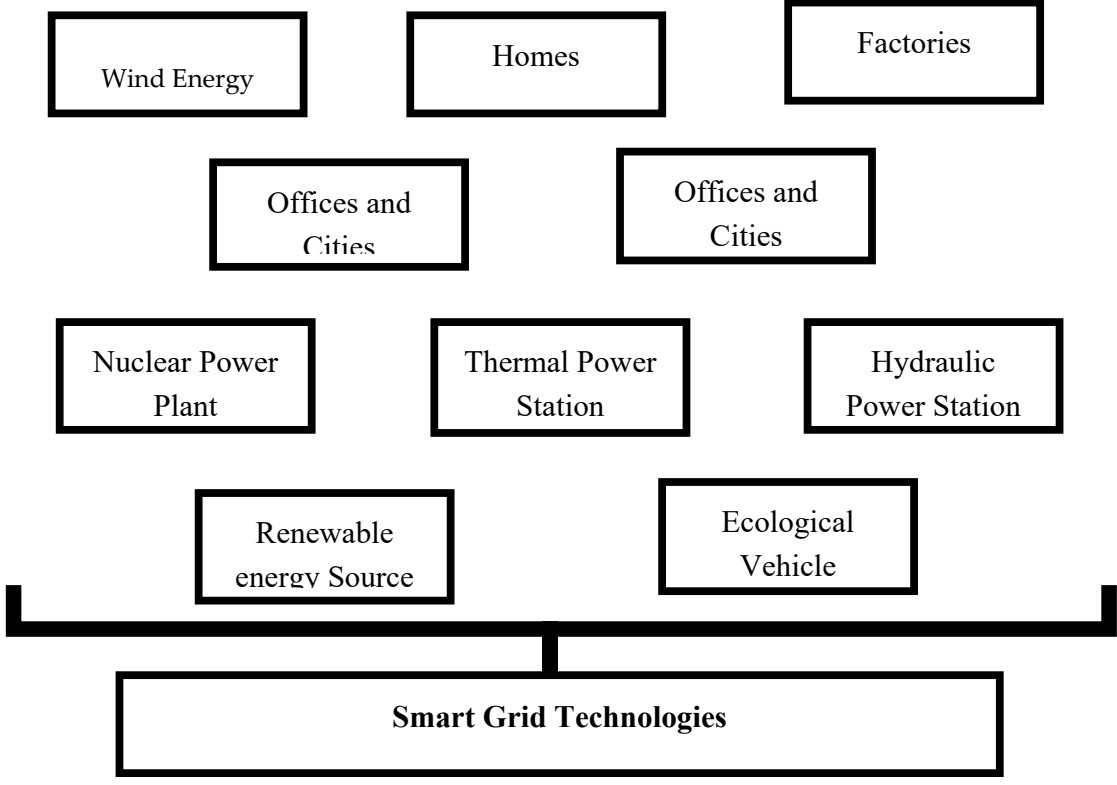

**Figure 2.** Block diagram of smart grid technology.

Such sources help people by generating electricity. The price of energy generated by coal and furnace oil is lower compared to electricity. Sound light power is considered cheaper and more resistant

to the use of solar power. Solar energy minimizes bills and makes people more comfortable and committed to the use of solar power during the day in place of furnace-based energy or coal-powered electricity. The smart grid can be described as a smart electrical grid integrating an electrical network with smart digital communication technologies. A smart grid has the potential to generate renewable electricity from a wide variety of wind turbines, solar energy projects, and even plug-in hybrid electric vehicles. Smart devices are able to determine how much energy they consume based on the preset preferences of their customers. It will contribute to a decrease in peak loads that impact prices for the production of electricity.

For example, clever sensors such as a sensor for thermal stations are used to control the temperature of the boiler based on predefined heat levels. The smart meters offer two-way communication between power suppliers and end-users to automate collection of billing data, to detect device failures, and to dispatch repair crews more quickly to the exact location. Smart substations are often required to break the flow path into several directions. Substations need massive and extremely expensive appliances to operate, including transformers, switches, condenser banks, disconnecting cables, and networking relays, among many others.

### 3.1. Smart Grid Technologies

A new technology called solar power was discovered about seven or eight years ago. This energy works on the principles of the sun's heat and strength. Throughout time, a new technology has been developed to minimize costs and to avoid problems of distortion resulting from the use of expensive electricity. Now, with the increasing financial and economic problems, a utility should provide a fair and cost-effective means of acquiring energy which customers can use easily and efficiently without charge. A few years ago, a new technology called smart grid launched. Networks are a set of wires that transfer electricity from one asset to another. It is an energy exchange between the receiver and the sender. It was developed in the 1960s, has since been improved, and gives customers and scientists a lot of wealth and ease regarding the safe and effective delivery of energy to their users.

### 3.2. Security Device

The user device initiates a request from WSN for service access to collect information. At the initial stage, the WSN registers the information of the user. The information, such as the device, identifies the physical address, and request time is registered by the WSN provider. The SG generates an authentication certificate with information such as IoT, CIT, CRSN, WSN, RDSRP HBO, and CBA. The tuples of the authentication certificate are explained in Table 1.

**Table 1.** Authentication certificate tuples and description.

| Tuple | Description |
|:-----:|:-----------:|
| IoT | Internet of Things |
| CIT | Communications and Information Technology |
| CRSN | Cognitive Radio Sensor Networks |
| WSN | Wireless Sensor Networks |
| RDSRP | Robust Bio-Dynamic Stimulated Routing Procedure |
| HBO | Hybrid Bird Optimizer |
| CBA | Clustering Based Approach |

### 3.3. Inspired Bio-Computing Model

The IBC (Inspired Bio-Computing Model) guides the development of new science and technology algorithms, methods, and strategies for solving complex and real problems. There are several bio-inspired approaches, such as biological, social, and mating algorithms. The IBC algorithm is an ideal solution for complex problems using a perfect balance between the components in the smart grid structure.

### 3.4. Optimization Using Butterfly Mating

Male butterflies look for the right color, weight, dimensions, and ventral surface of wings for female butterflies, which play key roles in cultivating early male pairing reaction. Ultraviolet patterns in butterfly wings allow male species to be identified. The females use ultraviolet models in which the males show the differences in their living conditions. The best receives more ultraviolet compared with the worst, which can be seen numerically as in Equation (1):

$$AB_{j \to i} = AB_j * \frac{E_{ji}^{-1}}{\sum_w E_{jw}^{-1}} \tag{1}$$

where i = 1, 2 ... m; j = 1, 2 ... n; and w = 1, 2 ... p. A $B_{j \to i}$ is the pattern absorbed by ultraviolet by ith female butterfly from the jth male butterfly. $AB_j$ is the emitted ultraviolet pattern by the jth male butterfly. $E_{ji}^{-1}$ is the female butterfly's Euclidean distance between the jth male and the ith factor. $E_{jw}^{-1}$ denotes the female butterfly's Euclidean distance between the jth male and the wth factor.

The crossover and mutation operators are integrated into the optimization phase during the assembly process to create an array of solid butterflies in the environment. A variant of the binomial crossover ($R_c$) and mutation (Tu) operators used in this analysis that is automatically updated is numerically displayed as in Equation (2):

$$CD_j^{h_w+1} = CD_i^{h_w+1} * (1 - R_c) + CD_i^{h_w} * R_c \tag{2}$$

where $h_w$ = generation current number and $CD_j^{h_w}$ = |th butterfly fitness.

From the above Equation (2), the $R_c$ crossover binomial is determined in the following Equation (3):

$$R_c = 0.9 + 0.1 * \frac{d\left(CD_j^{h_w}\right) - d\left(CD_{j,Best}\right)}{d(CD_{w,worst}) - d(CD_{j,Best)}} \tag{3}$$

where $CD_{j,Best}$ = best butterfly individual in the butterfly population and $CD_{w, worst}$ = best butterfly individual in the butterfly population. Mutation (Tu) operators used in the study analysis are updated in the following Equation (4):

$$Tu = CD(\overline{y} =)_i^{h_w+1} = \begin{cases} y_j^{h_w} + \nabla(h_w y_j^{\gamma} - y_j^{h_w}) \text{ if } \alpha \leq 0.4 \\ y_j^{h_w} - \nabla(h_w y_j^{\alpha} - y_j^{h_w}) \text{ otherwise} \end{cases} \tag{4}$$

where $\overline{y}$ = Female butterfly chromosome mutated genomic value and $\gamma, \alpha$ = random number uniformly distributed as (0, 1). From the above Equation (4), $\nabla(h_w, x)$ is determined in the following Equation (5):

$$\nabla(h_w, x) = x \left(1 - \gamma^{(1 - \frac{h_w}{h_{wmax}})^2}\right) \tag{5}$$

where i = 1, 2 ... m; j = 1, 2 ... n; and w = 1, 2 ... p. $h_{wmax}$ is the generation habitat maximum number, and x is the assignment binary value. It increases the probability of a combination, which can lead to greater fertility, longevity, and egg weights. After genomic depiction is done, the female butterfly places eggs on the gentian plants, resulting in a caterpillar population.

As inferred from Algorithm 1, in these phases, after receiving the initial messages from all sensor nodes in the area of interest, the initial population is generated by an altered number generator from 0 to 1 to solve the problem. Here, $\gamma_{aj}^m$ is determined as monogamous, $\gamma_{gj}^m$ is determined as polygynous, and $\gamma_j$ is the bird population. A group of individuals consists of one population that provides a complete solution to a problem identified in each person, represented by a 0-s or 1-s series. Genetic algorithms (GA) generation and static GA are the two most common methods used in initialization

processes for creating a new population of individuals. It results in a new population of current individuals and several previous generation individuals due to fusion and mutation to optimize solutions in the problem search area.

---

**Algorithm 1**

---

Initialization of the bird population using robust bio-dynamic stimulated routing procedure.
Procedure: determining the Bird Population Using Robust Bio-Dynamic Stimulated Routing Procedure
Input: $\gamma_j$
Execute the random set of population birds in an order.
Calculate $\gamma_j = \gamma_1, \gamma_2, \gamma_3 \ldots \gamma_m$
Fitness values are calculated till it attains $\gamma_j = \gamma_j^1 + \gamma_j^2 + \gamma_j^3 \ldots + \gamma_j^m$
Calculate monogamous $\gamma_j = \gamma_{aj}^1 + \gamma_{aj}^2 + \gamma_{aj}^3 \ldots + \gamma_{aj}^m$
Calculate polygynous $\gamma_j = \gamma_{gj}^1 + \gamma_{gj}^2 + \gamma_{gj}^3 \ldots + \gamma_{gj}^m$
Update $\gamma_j$
Finally, determine the $\gamma_j$
End for
Until repeat for all the coefficients determines the current individuals and several previous generation individuals due to fusion and mutation to optimize solution in the problem search area.
End procedure

---

Such caterpillars grow and have environmental degradation and extremes that lead to densities in a new habitat in the first few generations. In the end, at least the strongest caterpillar in the newly established population can be numerically shown in the following formula in a habitat (Equation (6)):

$$PC_{j,Best}^{hw+1} = \left\{ \begin{array}{c} PC_{j,}^{hw+1} d\left(PC_{j,}^{hw+1}\right) < d\left(PC_{i,}^{hw}\right) \\ PC_{i,}^{hw} \text{ otherwise} \end{array} \right\} \tag{6}$$

where $PC_{j,}^{hw+1}$ = fitness of the jth caterpillar, $PC_{i,}^{hw+1}$ = fitness of the ith caterpillar, and $PC_{j,Best}^{hw+1}$ = survival of the next-generation caterpillar. To use a competition feature DA for ensuring one caterpillar per bud, ∋ remains contest per competition, which can be seen numerically seen in the following Equation (7):

$$h_w\left(T_n, I_p\right) = \frac{DAd_j(\ni_i)T_n.I_p(DA_i)d_{\ni i}PC_j}{w_{pcj(DA_i)}} \tag{7}$$

where $T_n$ = population of the adult butterfly, $I_p$ = total number of plants, and $DAd_j$ = number of female butterfly eggs laid divided by two by assigning a 1:1 sex ratio. Therefore, the smallest population of an adult butterfly is denoted in the following Equation (8):

$$w_{pcj(DA_i)} = DAd_j(\ni_i)d_{\ni i}PC_j * \frac{T_n}{I_p}PC_j \tag{8}$$

Young caterpillars grow despite generalist predators and harsh conditions, and the other caterpillars are adult butterflies. The strongest butterfly is selected for each cycle, which after a certain time in a habitat can be further subjected to the maturing process and is shown numerically in the following Equation (9):

$$CD_{j,Best}^{hw+1} = \left\{ \begin{array}{c} CD_{j,}^{hw+1} d\left(CD_{j,}^{hw+1}\right) < d\left(CD_{i,}^{hw}\right) \\ CD_{i,}^{hw} \text{ otherwise} \end{array} \right\} \tag{9}$$

Each cycle that makes eggs into larvae and eventually into a pupa produces a lovely butterfly until the combination season ends. (As inferred from Algorithm 2).

---

**Algorithm 2**

---

Mating process using robust bio-dynamic stimulated routing procedure.
Procedure: determining Mating Process Using Robust Bio-Dynamic Stimulated Routing Procedure
Input: $D_j^{max}$- $D_j^{min}$
Determining the energy of available individual birds.
Calculate $\gamma_{Dj}$ = rand (.) $*D_j^{max}$- $D_j^{min}$
Male successful mate with female birds are calculated till it attains $\gamma_{Pb} = \exp(g\gamma^j\text{-}g\gamma^{n(j)})$
Calculate superior values $\gamma_j = \sqrt{\left(g_{Dj}^1 - g_{Dj}^1\right)^2 + \left(g_{Dj}^2 - g_{Dj}^2\right)^2 + \left(g_{Dj}^3 - g_{Dj}^3\right)^2 + \ldots \left(g_{Dj}^m - g_{Dj}^m\right)^2}$
Calculate polygynous $\gamma_{jBest} = \gamma_{gj}^1 + \gamma_{gj}^2 + \gamma_{gj}^3 \ldots + \gamma_{gj}^m$
Update $\gamma_{jBest}$
Finally, determine the $\gamma_{jBest}$
End for
Until repeat for all the whole process repeats until the minimum value of every bird energy factor is reached.
End procedure

---

At first, multiple birds are placed over these females during the mating process. Each bird's transformation is based on its appeal. When the allure is higher, the search space can be vast, which means that the partner bird alone has a flipping chance of the individual gene. Here $D_j^{max}$-$D_j^{min}$ is the available energy of birds, $\gamma_{Pb}$ is the probability of the male bird population, and g denotes the female bird. The simulated randomizing approach has been utilized for selecting the best male in a random population with the females in each stage in the search area. When the approximate likelihood was compared, a random number between 0 and 1 was produced. If the risk is less than the likelihood, males successfully pair the female with their genome; otherwise, they are discarded in the female's egg. This bio-inspired technique has been integrated into the components during data transmission. The matched data has been processed using the intelligent grid, which is the system integrating advanced sensing, control, and integrated communications systems in the current electricity network shown in Figure 3, where sender and receiver nodes represent male and female birds. In terms of intelligent central generation, intelligent transmission, intelligent substation, intelligent distribution feeders, and smart metering, the smart grid is incomparable to the current grid. Two-way communication interoperability between advanced applications are reliable and safe. Also, low-response communications and sufficient bandwidth robust network protection avoid cyber-aggression in network stability and security with sophisticated controls.

Addressing, the rising world demand for power, reducing the cost of global power upheavals and failure networks, reducing $CO_2$ emissions. By improving green power generation and energy consumption, avoiding raising electricity prices by regulating demand and supply. Providing long-term reliable services, and replacing aging infrastructure and employees are important for smart grids. During the data collection process, both transmission and distribution smart grid processing of the observed data are carried out from the sensor nodes. First, it is up to the client to initiate the process of data collection by frequently sending to sensor nodes a predefined number of data collection messages through the base point and sinks within the network, as shown in Figure 4.

In an intelligent grid environment, the sensor nodes are explored, decoding data messages correctly. Each sensor node then transfers the secret code to the nearest node sensor before the packets are transmitted through the system.

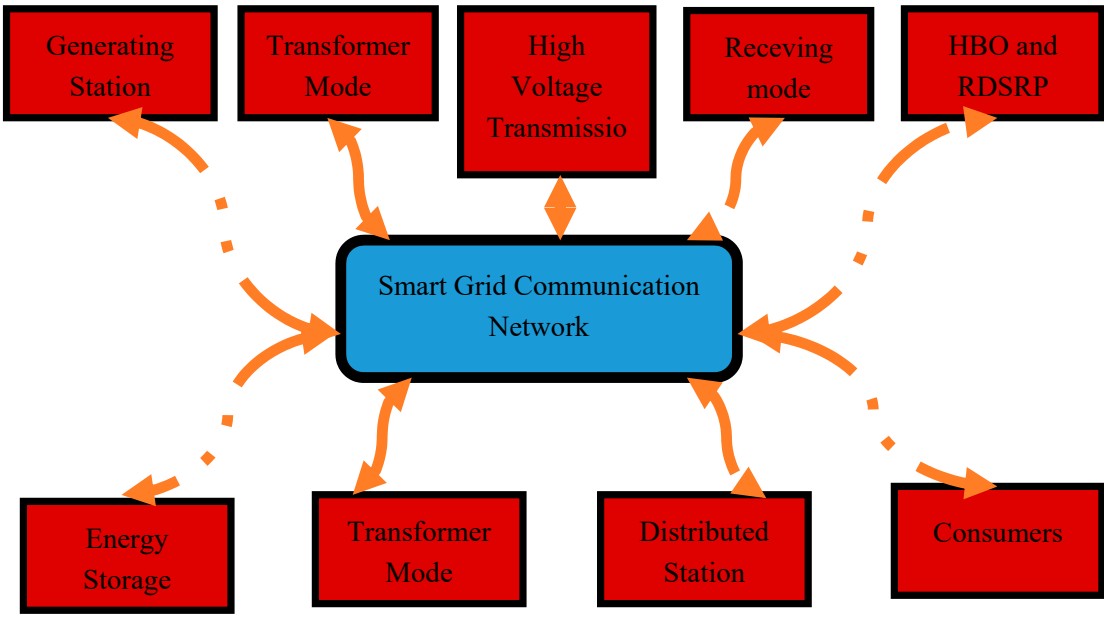

**Figure 3.** Schematic block diagram for smart grid systems.

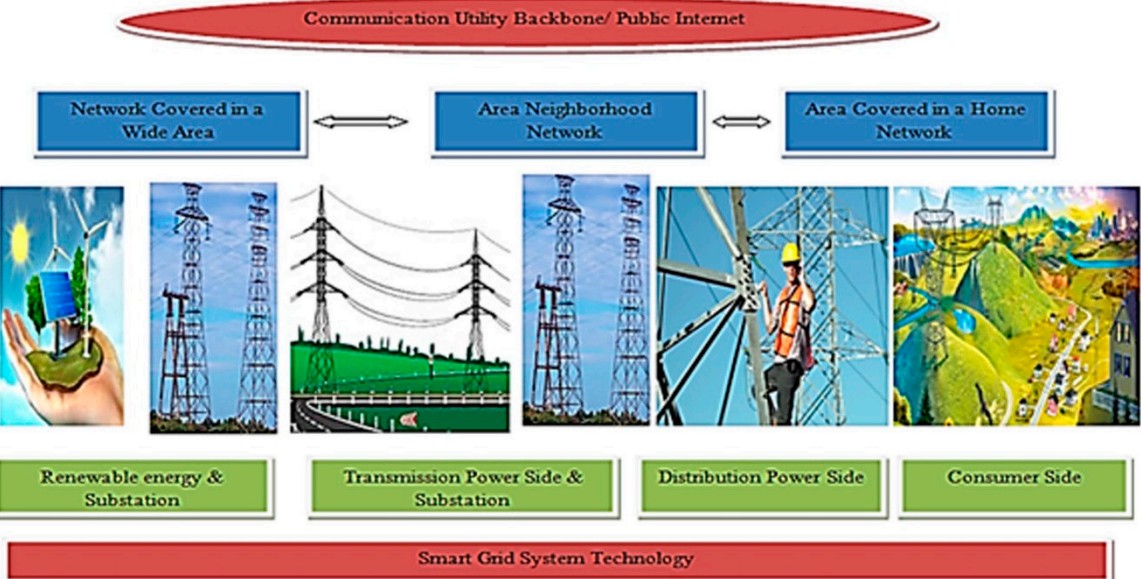

**Figure 4.** Wireless sensor network survey for smart grid systems.

If valid data receives the free node sensor channel, the following data will be transmitted via the network. It tracks the environment continuously, saves information in the cache, and awaits a predefined time for channel acquisition. This information is then moved directly to the corresponding sink when the media is obtained.

In this context, it is important to notice that the collected data are sent directly to the associated sink if the sensor nodes are in the transmission range. The data sent from the sensor node cache is immediately removed after transmission of the event-related information to allow the memory to be updated for new events. Given the history stored in the network, the next transmission deadline for the sensor node is immediately regularly updated. Therefore, the collection of the objective information is determined by variable $\varphi$, and it is numerically explained in Equation (10):

$$\varphi = \min CD_{j,Best}^{h_w+1} + \max PC_{j,Best}^{h_w+1} \tag{10}$$

The main purpose is to reduce the total cost of data acquisition for the surveillance of network intelligent grid events. The residual packet energy is subjected to the throughput of the network, and it is denoted in the following Equation (11):

$$\sum\nolimits_{d_j \ni d_m} y_{d_j}\, d_w \geq 1 \tag{11}$$

Minimum skin network is deployed in the guaranteed smart grid environment systems, and it is described in the following Equation (12):

$$S_i R_j \ni T_w \;=\; 1 \tag{12}$$

where $S_i$ = sensor nodes, $R_j$ = monitoring events in smart grid systems, and $T_w$ = random skin network.

From the above Equation (12), the sensor region denotes the smart grid skin environment, and it follows Equation (13):

$$y_{d_j} \leq y_{d_j} d_w \;=\; 1 \tag{13}$$

In this formula, the packet data probability moving from the source to the destination via the j-data path is shown in Equation (14):

$$\varphi y_{d_j} = \sum\nolimits_{d_j \ni d_m} y_{d_j} \frac{1}{\sum_{d_j \ni d_m} y_{d_{j^*} \frac{T_n}{I_p} PC_j}} \tag{14}$$

The sum of the datasets collected from the sink and the base station is the same as those sent from the source sensor $S_s$ through a sequence of sensor nodes $S_n$ along with the network; the routing paths follow Equation (15):

$$\sum\nolimits_{S_s \ni S_n} y_{d_j} = \frac{1}{\sum_{d_j \ni d_m} y_{d_{j^*} \frac{T_n}{I_p} PC_j}} \tag{15}$$

Data path loops during network transmission data are avoided by middle sensors, and thus, the intermediate routes are determined in the following Equation (16):

$$\sum\nolimits_{S_s \ni S_n} y_{d_j} > \sum\nolimits_{d_j \ni d_m} y_{d_j} \tag{16}$$

More packet data are transmitted with the help of a normal sensor network, and it ensures that the guaranteed smart grid systems follow Equation (17):

$$\sum\nolimits_{S_s \ni S_n} y_{d_j}\, (S_n + 1, S_s + 1) \;\leq\; 1 \tag{17}$$

A collection of the sensors allocated should be more than the total buffer $T_b$ size as the node network and the amount of the data packets obtained from each sensor.

$$T_b = (S_n + 1, S_s + 1) \;\leq \ni_0 \tag{18}$$

The energy consumption position of a sensor node in a region shall not surpass its original network capacity $N_c$, and the total collection of the network is determined as follows in Equation (19):

$$N_c = \min \frac{1}{S^2} \int_0^S \ni_1 (s) w(y,s)\, dx \tag{19}$$

Log-of-zero is a negative limit, while the utility exponent is null on the network for an inaccessible alternative skin of the network, and it is denoted in the following Equation (20):

$$N_c = \sum\nolimits_{S_s \ni S_n} y_{d_j} * \frac{1}{\frac{T_n}{I_p} PC_j} \tag{20}$$

The total distance from the corresponding networking sensors and power consumption to the sink are determined. A neighboring node in the SG can be found by each sensor node, new or inactive. Two-way QoS communication interferes with SG network-based applications for wireless sensors. Hence, for the smart system 4.0 framework, a highly reliable communication network based on the WSN is critically important for successful operation of the electricity grids in the next decade.

## 4. Results and Discussion

The performance of the proposed Robust Bio-Dynamic Stimulated Routing Procedure (RDSRP) based on the real-time behavior of new Hybrid Bird Optimizer (HBO) model smart grid wireless sensor applications is evaluated using a WSN communication environment modeled in an opportunistic network simulator with 400 mobile wireless devices. The devices are interconnected with a WSN for stored information (dataset and images) retrieval. Table 2 describes the detailed simulation setting for evaluation.

**Table 2.** Simulation setting.

| Parameter | Configuration Value |
|---|---|
| Wireless Devices | 400 |
| Request Size | 68 bytes |
| Number of Infrastructure Units | 20 |
| Bandwidth | 4 Mbps |
| Request Expiry Time | 250 ms |
| Signature Size | 300 bits |
| Time Slots | 350 |

Figure 5 shows that each routing scheme provides a client with a maximum number of data packets sent to real-time SG-monitoring events. It clearly shows that the PDR (Packet Delivery Ratio) is growing by increasing the network size from 100 to 500 in the two routing schemes. As sensor nodes begin to die between 400 and 500 SG in the final rounds, the PDR decreases rapidly. The decreasing rates of PDR in FBR, IIP, ISWN, CRSN, an CBA are more than that of RDSRP and HBO in SG. The PDR of RDSRP and HBO depends on data traffic load for the provided events and offers information of about 97% of the simulation studies showing both routed systems. The PDR is based on a relatively stable, less congested path choice in which significant residual energy is retained in the SG in the variable sensor nodes.

Moreover, the intermediate nodes along an itinerary depend on its number as a result of the overloading number of nodes, shown in Figure 6a,b, increasing the likelihood of memory overrun problems and incomplete data. This processing overload leads to problems with congestion control for sensor nodes.

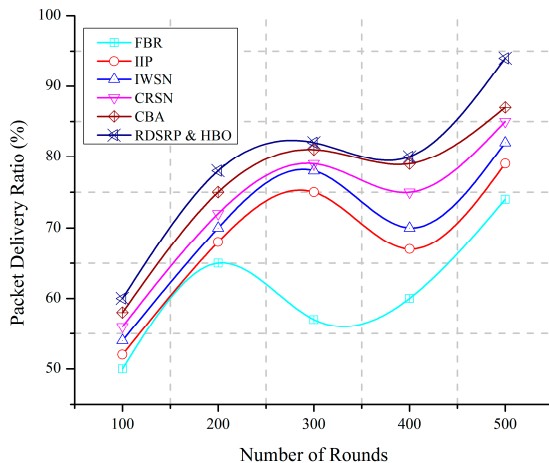

**Figure 5.** Packet delivery ratio.

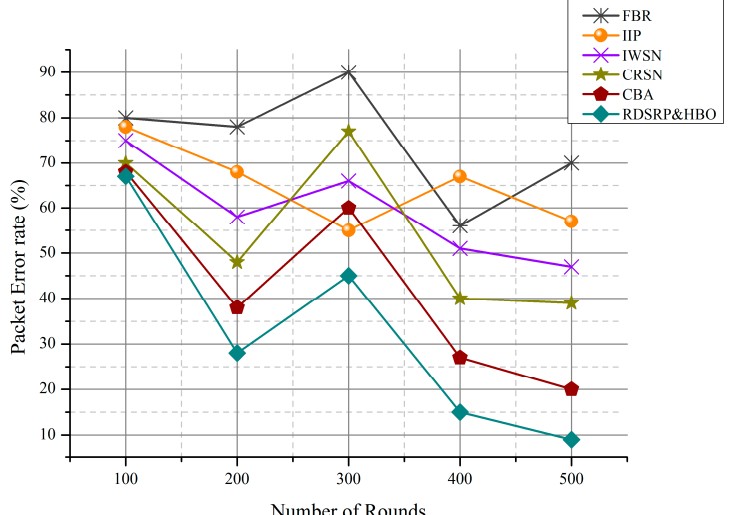

**(a)** Packet Error Rate.

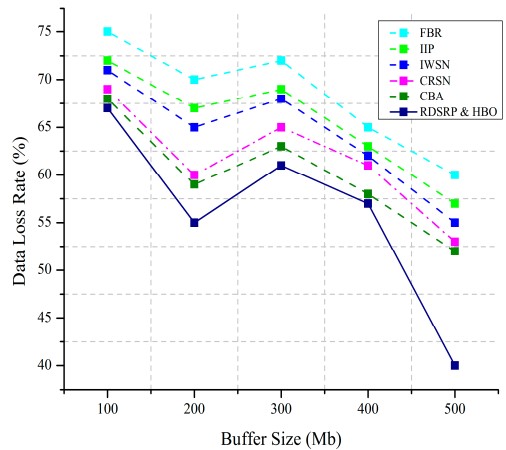

**(b)** Data Loss Rate

**Figure 6.** Overloading Comparison.

Initially, the PDR performances of FBR, IIP, ISWN, CRSN, and CBA in the smart grid are small. The probabilities of FBR, IIP, and ISWN, are, however, greater than that of CRSN and CBA routing schemes between 100 and 300 rounds, and similar activity exists between 300 and 500 rounds. It as a result of this that the PDR performances of FBR, IIP, ISWN, CRSN, and CBA are less than that of the proposed RDSRP and HBO. This leads to a high RDSRP and HBO network throughput in the smart grid, as shown in Figure 7.

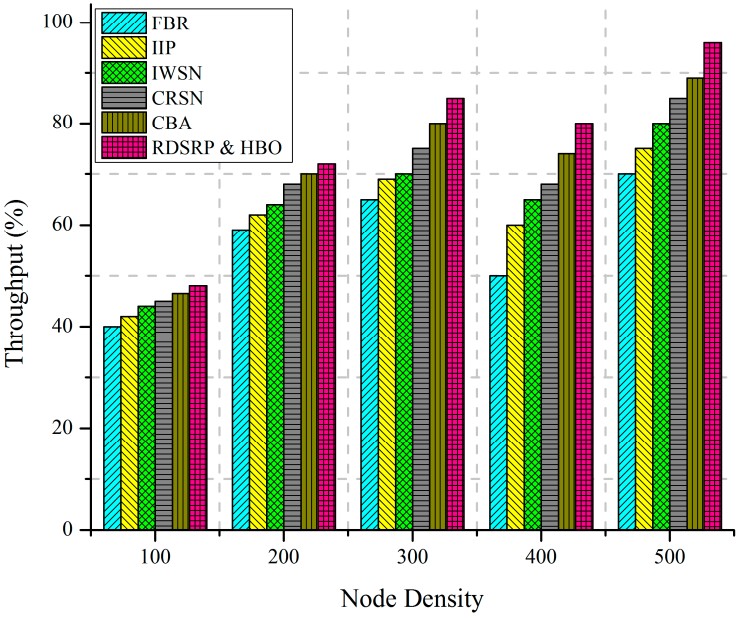

**Figure 7.** Throughput.

Table 3 gives a detailed analysis of the network delay model. It is obvious that, when moving packets from the source to the destination in SG, the routing scheme of RDSRP and HBO is less delayed than FBR, IIP, ISWN, CRSN, and CBA. Efficient transmission over the chosen routing route for tracking events will be mostly improbable in FBR, IIP, ISWN, CRSN, and CBA according to the intermediate hop counts. Here, let us assume that some packet numbers will be invalid because a threshold of the origin to the network sink does not meet a given period. In simulation tests, the length of the path caused by many intermediate slopes has increased for comparison with RDSRP and HBO in the SG. When looking at the simulation facts, it is shown that our scheme uses the nearly optimal shortest paths to the sink in the network from a source. Therefore, compared to FBR, IIP, ISWN, CRSN, and CBA for SG monitoring instances, the designed system (RDSRP and HBO) leads to low network delays, as shown in Table 3.

**Table 3.** Network delay.

| Number of Rounds | FBR | IIP | IWSN | CRSN | CBA | RDSRP and HBO |
|:---:|:---:|:---:|:---:|:---:|:---:|:---:|
| 100 | 75.6 | 74.3 | 72.8 | 71.1 | 70.9 | 70 |
| 200 | 67.8 | 65.3 | 63.9 | 62.3 | 60.9 | 59.6 |
| 300 | 70.7 | 68.4 | 64.5 | 60.2 | 58.6 | 54.5 |
| 400 | 65.6 | 60.6 | 59.8 | 57.6 | 54.6 | 50.8 |
| 500 | 60.5 | 57.4 | 55.4 | 50.3 | 46.3 | 36.5 |

However, when the networks range from 100 to 500 capsules, the low late performance of HLR-AODV (Health, Link Quality and Reputation Aware Routing Protocol) is greater equated to smart

grid ETL-AODV routing. Here, we noted that the delay in RDSRP and HBO's small, medium, or large networks is excellent compared to the FBR, IIP, ISWN, CRSN, and CBA smart grid routing schemes. This is because of its complex network adaptability, as optimal routes are identified towards the sink. Further, when sending data or requesting packet to the next-hop node, memory overrun issues are properly considered compared with FBR, IIP, ISWN, CRSN, and CBA routing schemes, as shown in Figure 8.

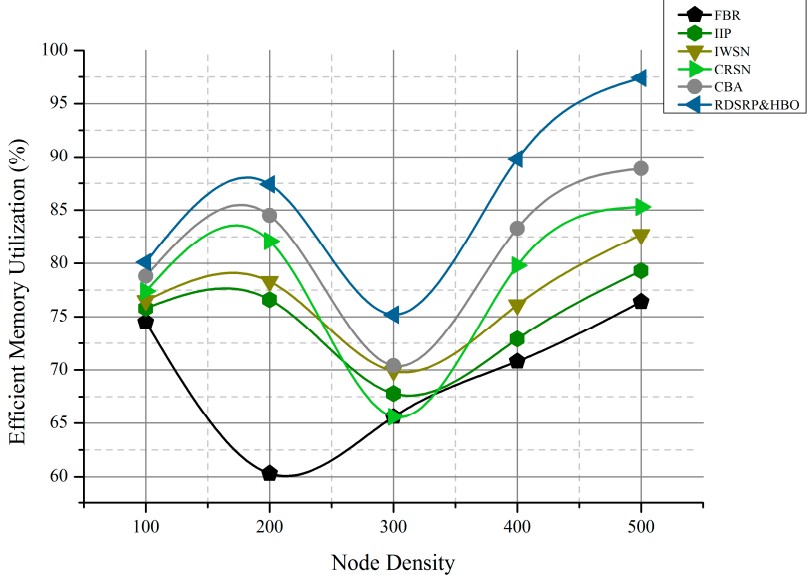

**Figure 8.** Efficient memory utilization.

A detailed view of the FBR, IIP, ISWN, CRSN, CBA, and RDSRP and HBO methods' residual energy profile is given in Table 4. This shows clearly that the RDSRP and HBO energy residual profile is higher than the FBR, IIP, ISWN, CRSN, and CBA methods in the SG. Such low power consumption leads to a long-lasting network service life for the RDSRP and HBO method in the SG. Based on the sensor node location, the system is designed to split the whole sensor network into two main groups, which minimizes sensor, transmission data buffer, overflow latency, and data traffic loads to both ends.

**Table 4.** Residual energy.

| Number of Rounds | FBR | IIP | IWSN | CRSN | CBA | RDSRP and HBO |
|:---:|:---:|:---:|:---:|:---:|:---:|:---:|
| 100 | 74.5 | 75.8 | 76.5 | 77.4 | 78.8 | 80.1 |
| 200 | 60.3 | 76.6 | 78.3 | 82.1 | 84.5 | 87.4 |
| 300 | 65.6 | 67.8 | 69.9 | 65.6 | 70.4 | 75.2 |
| 400 | 70.8 | 72.9 | 76.1 | 79.8 | 83.3 | 89.8 |
| 500 | 76.4 | 79.3 | 82.7 | 85.3 | 88.9 | 97.4 |

Therefore, these methods cannot completely optimize the quality of the data path for successful data transmission. Moreover, in most of these schemes, there is an efficient neighborhood discovery mechanism important to robust routing. The loss leads to an unnecessary delay when a relay node dies on the path in the network. Figure 9a shows the number of nodes that died per number of rounds (100–500). Figure 9b shows the number of nodes that died per number of rounds (600–1000).

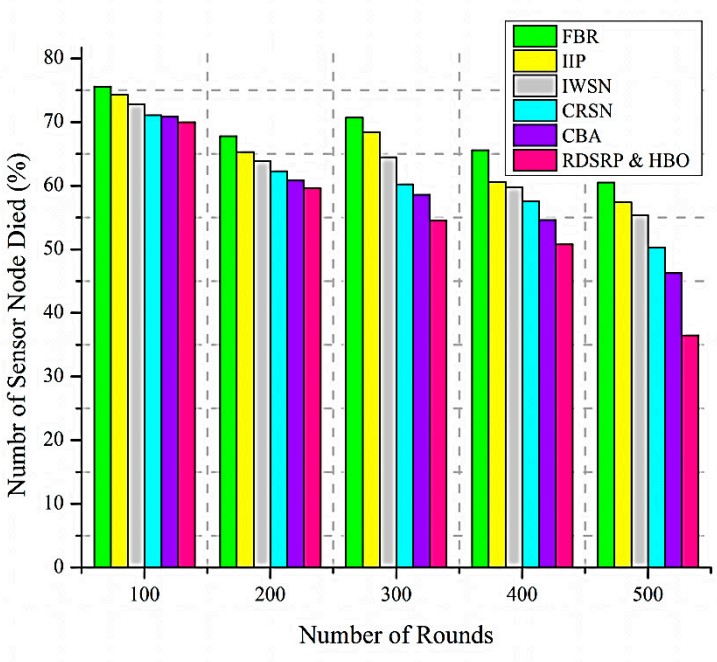

**(a)** Nodes that died vs. rounds (100–500).

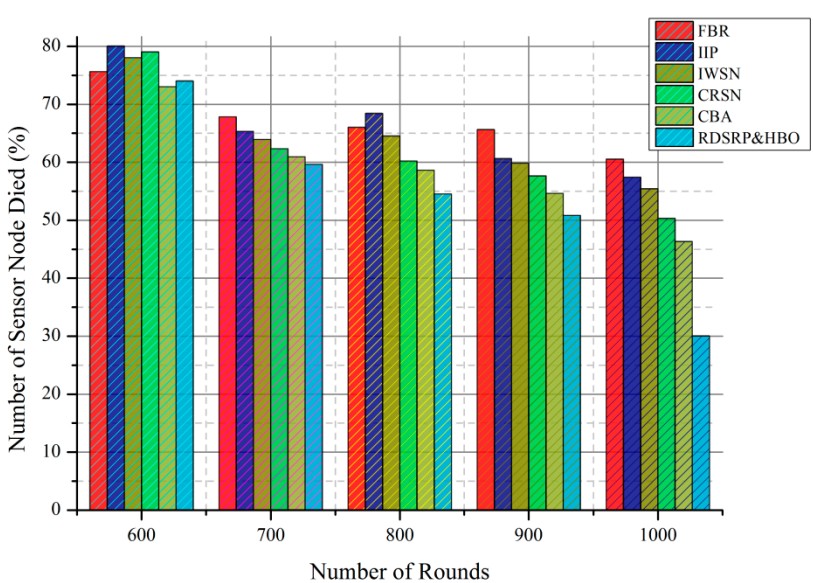

**(b)** Nodes that died vs. rounds (600–1000).

**Figure 9.** An efficient neighborhood discovery mechanism.

From the experimental results, it was shown that the above problem uses the Robust Bio-Dynamic Stimulated Routing Procedure (RDSRP) based on the real-time behavior of a new Hybrid Bird Optimizer (HBO) model. The proposed distributed system significantly improves network efficiency and reduces the transmission of excess packets for WSN-based SG applications.

The current study explores some of the most recent advances in this area, including energy efficiency and energy harvesting, which are expected to have a significant impact on green topics under smart systems the Internet of things. The experimental results show that the proposed distributed

system suggestively enhances network efficiency and reduces the transmission of excess packets for wireless sensor network-based smart grid applications.

## 5. Conclusions and Future Outcomes

The traditional electricity grids are converted into SG, inspired every day by the 4th industrial revolution. To this end, the main communication technologies promising for multiple SG applications are IoT and IWSNs that allow the smart system 4.0 framework. This hampers the two-way QoS communications specifications for SG-based wireless sensors. To address these issues, this paper proposes a Robust Bio-Dynamic Stimulated Routing Procedure (RDSRP) based on the real-time behavior of new Hybrid Bird Optimizer (HBO) model smart grid wireless sensor applications. The bio-inspired routing system offers an extremely reliable multiple-hop communication networking architecture in a smart system 4.0 framework for the SG applications of WSN-based QoS-aware information collection. The researchers will create an innovative interaction system to ensure various QoS-conscious data collection with minimal redundancies for different WSN-based SG applications. Therefore, an important future research path is the experimental evaluation of the procedures proposed in WSN testbeds.

**Author Contributions:** Conceptualization, J.S. and X.C.; methodology, S.K.; software, R.S.; validation, S.K., X.C., and J.S.; formal analysis, J.S.; investigation, R.S.; resources, X.C.; data curation, X.C.; writing—original draft preparation, X.C.; writing—review and editing, S.K.; visualization, R.S.; supervision, X.C.; project administration, X.C.; funding acquisition, X.C. All authors have read and agreed to the published version of the manuscript.

**Funding:** This work was sponsored in part by National Natural Science Foundation of China (71502072).

**Conflicts of Interest:** The authors declare no conflict of interest.

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
