# Peer review of "Internet-of-Things-Assisted Smart System 4.0 Framework Using Simulated Routing Procedures"

_sustainability, doi:10.3390/su12156119_

Round 1
Reviewer 1 Report
The manuscript sustainability-826549 reports the applications of the Internet of things and Wireless Sensor Networks to enhance the ability of smart grid information in Smart System Frameworks.
This work presents a novel approach to improve the reliability of Wireless Sensor Networks by the development of the so-called Robust Bio-Dynamic Stimulated Routing Protocol.
Similarly, Wireless Sensor Network based Smart Grid applications are presented to confirm experimental results. The work also provides the evaluation of the proposed method using a simulation setting. However, the presentation of experimental results could be improved by including the clarification of several aspects. For these reasons, it is recommended the revision of the present version before publication.
Abstract:
- The relevance of experimental results as well as a conclusion consistent with the results could improve the impact of the research.
- The term ‘Smart grid’ is first introduced in Line 17, although the acronym appears in line 20.
Section 3. Robust Bio-Dynamic Stimulated Routing Protocol (RDSRP) based on the real-time behavior of 122 a new Hybrid Bird Optimizer (HBO) model:
- The acronym ‘QoS’ is first introduced in Line 59, although its definition appears in line 106.
- The meaning of the following sentences needs to be clarified since it may lead to misinterpretation (Lines 130-132) ‘The price of energy generated by coal and furnace oil is lower compared to electricity. Sound light power is considered to be cheaper and more resistant to the use of solar power’.
- Use the same term ‘bio- inspirited’ in Line 64 or ‘bio-inspired’.
- Revise punctuation errors in this sentence (Line 265-268): ‘Addressing the rising world demand for power Reducing the cost of global power upheavals and failure networks reducing CO2 emissions by improving green power generation and energy consumption avoid raising electricity prices by regulating demand and supply Providing long-term reliable services, replacing aging infrastructure and employees’.
- Figure 4: the quality of the image could be improved, the picture is stretched or distorted.
Results and discussion: The presentation of the results may be improved by discussing the consistency of data. For instance, the criteria for selection of the simulation setting variables or the routing schemes. Additionally the contribution of the proposed method over previous works, including strengths and weaknesses should be considered.
Typo and grammar errors:
Line 52: efficiently and efficiently.
Line 59-60: revise the sentence: ‘Due to the security and complex QoS specifications of different factories and the process of implementation is difficult [18]-[24]’
Line 74: lowercase letter before full stop ‘. section III discusses the proposed methodology with'.
Figure 2: revise ‘renuwable’, is incorrect.
Author Response
The manuscript sustainability-826549 reports the applications of the Internet of things and Wireless Sensor Networks to enhance the ability of smart grid information in Smart System Frameworks.
This work presents a novel approach to improve the reliability of Wireless Sensor Networks by the development of the so-called Robust Bio-Dynamic Stimulated Routing Protocol.
Similarly, Wireless Sensor Network based Smart Grid applications are presented to confirm experimental results. The work also provides the evaluation of the proposed method using a simulation setting. However, the presentation of experimental results could be improved by including the clarification of several aspects. For these reasons, it is recommended the revision of the present version before publication.
Abstract:
- The relevance of experimental results as well as a conclusion consistent with the results could improve the impact of the research.
Response:updated and verified
- The term ‘Smart grid’ is first introduced in Line 17, although the acronym appears in line 20.
Response: updated and verified
Section 3. Robust Bio-Dynamic Stimulated Routing Protocol (RDSRP) based on the real-time behavior of 122 a new Hybrid Bird Optimizer (HBO) model:
- The acronym ‘QoS’ is first introduced in Line 59, although its definition appears in line 106.
Response: it has been updated according to the corresponding lines.
- The meaning of the following sentences needs to be clarified since it may lead to misinterpretation (Lines 130-132) ‘The price of energy generated by coal and furnace oil is lower compared to electricity. Sound light power is considered to be cheaper and more resistant to the use of solar power’.
Response:The price of energy due to coal and furnace oil is minimized when compare to solar power.
- Use the same term ‘bio- inspirited’ in Line 64 or ‘bio-inspired’.
Response: updated.
- Revise punctuation errors in this sentence (Line 265-268): ‘Addressing the rising world demand for power Reducing the cost of global power upheavals and failure networks reducing CO2 emissions by improving green power generation and energy consumption avoid raising electricity prices by regulating demand and supply Providing long-term reliable services, replacing aging infrastructure and employees’.
Response:Addressing the world demand for the cost of global power and failure networks, Reducing CO2 emissions by improving green power generation and energy consumption helps to avoid raising electricity prices by regulating power demand
- Figure 4: the quality of the image could be improved, the picture is stretched or distorted.
Response: It has been updated
Results and discussion: The presentation of the results may be improved by discussing the consistency of data. For instance, the criteria for selection of the simulation setting variables or the routing schemes. Additionally the contribution of the proposed method over previous works, including strengths and weaknesses should be considered.
Response:The consistency of the data PDR is growing by increasing the network size from 100 to 500 in the two routing schemes. As sensor nodes begin to die between 400 and 500 SG in the final rounds, the PDR decreases rapidly.
Typo and grammar errors:
Line 52: efficiently and efficiently.
Response: It has been updated
Line 59-60: revise the sentence: ‘Due to the security and complex QoS specifications of different factories and the process of implementation is difficult [18]-[24]’
Response:The process of implementation of the factories are difficult due to the security and complexity of the Quality of Service (QoS) specifications
Line 74: lowercase letter before full stop ‘. section III discusses the proposed methodology with'.
Response: It has been updated
Figure 2: revise ‘renewable’, is incorrect.
Response: It has been updated
Reviewer 2 Report
Authors paper propose the Robust Bio-Dynamic Stimulated Routing Protocol (RDSRP) based on the real-time behavior of a new Hybrid Bird Optimizer (HBO) model in Smart Grid.
Overall, the author tried to draw conclusions well through experiments. However, this paper has some critical supplementations as follows.
- The title should be more specific.
- “Introduction” should be improved; The authors described the proposed system, but it is necessary to explain more specifically for the background.
- In Figures 1 is not expressed clearly to understand the structure of the proposed system. Also, in “Introduction”, the description of the Figure 1 should be clearly and specifically.
- As shown in Figures 2, the description of the “Smart Grid Technology“ is too general. In my opinion, it need to add a picture of the specific idea content rather than the general content.
- The paper needs the whole system flow chart or figures to understand clearly the flow of the whole proposed system.
- The overall Figures are very low quality. And the description of the Figure 3, 4 should be clearly and specifically.
- It is necessary to compare and analyze what the proposed system is different from the existing system.

Author Response
- The title should be more specific.
Response: Internet of Things assisted Smart System 4.0 Framework using simulated routing procedures
- “Introduction” should be improved; The authors described the proposed system, but it is necessary to explain more specifically for the background.
Response: The improved flexibility of the intelligent grid allows greater penetration of highly variable sources of renewable energy, such as solar power and wind power, even without energy storage. Current network technology is not intended for a vast number of dispersed feed-up points and typically the system at the receiving stage cannot support it, even though any feed-in is allowed at local (distributive) stage. Fast variations, for instance due to cloudy or rainy weather, are a major challenge for power engineers who need to ensure stable electricity by changing the outputs of more controllable generators, such as gas turbines and hydraulic generators. For this reason, intelligent grid technology is a necessary prerequisite for very large amounts of renewable energy in the grid.
- In Figures 1 is not expressed clearly to understand the structure of the proposed system. Also, in “Introduction”, the description of the Figure 1 should be clearly and specifically.
Reponses: As inferred from the Figure.1. Domain Access software is a software application which provides full life cycle access for remote management to the domain Operating system(OS) and the Base Management Card (BMC) in the data centre. Power cycling helps users to control servers remotely while they are disabled and to turn up / down or restart servers while they are operating the system OS based on web service systems. Cloud Access requires all on- and off-band domain control based on the protocol control panel. To address the hardware problems preventing the server from booting normally, access BIOS (base input/output system) or BMC directly from the device program
- As shown in Figures 2, the description of the “Smart Grid Technology “is too general. In my opinion, it need to add a picture of the specific idea content rather than the general content.
Response: Such sources help people by generating electricity. The price of energy generated by coal and furnace oil is lower compared to electricity. Sound light power is considered to be cheaper and more resistant to the use of solar power. Solar energy minimizes bills and makes people more comfortable and committed to the use of solar power a day in place of furnace-based energy or coal-powered electricity. The smart grid can be described as a smart electrical grid integrating electrical network with smart digital communication technologies. A smart grid has the potential to generate renewable electricity from a wide variety of wind turbines, solar energy projects and even plug-in hybrid electric vehicles. Smart devices are able to determine how much energy they consume based on pre-set preferences of their customers. It will contribute to a decrease in peak loads that impact prices for the production of electricity. For example, clever sensors such as a sensor for thermal stations used to control the temperature of the boiler based on pre-defined heat levels. The smart meters offer two-way communication between power suppliers and end-users to automate collection of billing data, detect device failures, and dispatch repair crews more quickly to the exact location. Smart substations are often required to break the flow path in several directions. Substations need massive and extremely expensive appliances to operate, including transformers, switches, condenser banks, disconnecting cables, networking relays, among many others
- The paper needs the whole system flow chart or figures to understand clearly the flow of the whole proposed system.
Response: it has been clearly mentioned in the Figure.3. and Figure.4
- The overall Figures are very low quality. And the description of the Figure 3, 4 should be clearly and specifically.
Response: The quality and pixel range has bene improved.
- It is necessary to compare and analyze what the proposed system is different from the existing system.
Response: All the results have been checked for consistency and numerically verified
Reviewer 3 Report
(1) Need overall process of RDSRP algorithm. Namely, algorithm 1 and 2 should connect each other and be showed as one RDSRP process.
(2) It is not clear why your proposed algorithm is better than other methods. Graphs showed high performance, but the detailed simulation process did not explained well. What are FBR, IIP, IWSN, CRSN and CBA's algorithms and simulation details? Block diagrams are good to explain for these methos. You explained these in "2. Leterature Survey", but simulation processes for these methods must be explained because the aim of this Journal must be provided experimental details so that the results can be reproduced.
(3) What is your new idea in algorithm 1 and 2? Many functions look like already existed ones.
Author Response
- Need overall process of RDSRP algorithm. Namely, algorithm 1 and 2 should connect each other and be showed as one RDSRP process.
Response: A group of individuals consists of one population that provides a complete solution to a problem identified in each person represented by a 0-s or 1s series. Genetic algorithms (GA) generation and static GA are the two common methods used in initialization processes for creating a new population of individuals. It results in a new population of current individuals and several previous generation individuals due to fusion and mutation to optimize solutions in the problem search area.
- It is not clear why your proposed algorithm is better than other methods. Graphs showed high performance, but the detailed simulation process did not explained well. What are FBR, IIP, IWSN, CRSN and CBA's algorithms and simulation details? Block diagrams are good to explain for these methos. You explained these in "2. Leterature Survey", but simulation processes for these methods must be explained because the aim of this Journal must be provided experimental details so that the results can be reproduced.
Response: It clearly shows that the PDR is growing by increasing the network size from 100 to 500 in the two routing schemes. As sensor nodes begin to die between 400 and 500 SG in the final rounds, the PDR decreases rapidly. The decreasing rate of PDR in FBR, IIP, ISWN, CRSN, CBA is more observed than that of RDSRP & HBO in SG. The PDR of RDSRP & HBO depends on data traffic load for provided events and offers information of about 97% of the simulation studies show that in both routed systems. This study explores some of the most recent advances in this area, including energy efficiency and energy harvesting, which are expected to have a significant impact on green topics under smart systems the Internet of Things. The experimental results show that the proposed distributed system suggestively enhances network efficiency and reduces the transmission of excess packets for Wireless Sensor Network based Smart Grid applications.
- What is your new idea in algorithm 1 and 2? Many functions look like already existed ones.
Response: Introducing bio inspired method is the novel technique and it has been shown in the Eq(1) to (5).
Reviewer 4 Report
The paper is interesting and has scientific value, but requires corrections.
It is written in a way that makes it difficult to understand the meaning of its individual fragments, especially in the initial part. It is necessary to rewrite the introduction, part 2 and the fragment of part 3 to better understand the purpose of the paper and the use of Bio-computing Model. A lot of information is included in the initial part, but it is not known what the article will be about: a specific algorithm or a broad analysis of the use of WSN for Smart Grid. In addition, please consider changing the title of the paper, as it mainly concerns the implementation and analysis of the effectiveness of a specific algorithm, and not a review of IoT and WSN applications in the Smart System 4.0 framework.
I would like to point out that the figures are of low quality. What's more, e.g. Fig. 1, it is not known whether this is the work of the authors or whether the graphics were taken from other sources.
The paper requires formatting in accordance with the requirements of the journal. In many places there are no punctuation marks, and the way of writing, e.g. variables from mathematical expressions is simply messy (for example, lines 163-166).
There is also a lack of proper quality and diligence. For example, in lines 158-186, the definition of parameter d cannot be found, and it is used in Equation 4.
Also in the case of Algorithm 1 and Algorithm 2 it should be clearly described how it works in the context of Smart Grid. Their presentation needs considerable improvement. It is suggested to present the operation of these algorithms in a graphic way that would facilitate their understanding for readers.
Author Response
The paper is interesting and has scientific value, but requires corrections.
It is written in a way that makes it difficult to understand the meaning of its individual fragments, especially in the initial part. It is necessary to rewrite the introduction, part 2 and the fragment of part 3 to better understand the purpose of the paper and the use of Bio-computing Model. A lot of information is included in the initial part, but it is not known what the article will be about: a specific algorithm or a broad analysis of the use of WSN for Smart Grid. In addition, please consider changing the title of the paper, as it mainly concerns the implementation and analysis of the effectiveness of a specific algorithm, and not a review of IoT and WSN applications in the Smart System 4.0 framework.
Response: Thank you for the comments rendered
I would like to point out that the figures are of low quality. What's more, e.g. Fig. 1, it is not known whether this is the work of the authors or whether the graphics were taken from other sources.
Response: Corresponding citation has been included in the context. The Quality of the figure has been improved.
The paper requires formatting in accordance with the requirements of the journal. In many places there are no punctuation marks, and the way of writing, e.g. variables from mathematical expressions is simply messy (for example, lines 163-166).
Response: it has been changed
There is also a lack of proper quality and diligence. For example, in lines 158-186, the definition of parameter d cannot be found, and it is used in Equation 4.
Response: it has been changed and mentioned it as best butterfly individual in butterfly population
Also in the case of Algorithm 1 and Algorithm 2 it should be clearly described how it works in the context of Smart Grid. Their presentation needs considerable improvement. It is suggested to present the operation of these algorithms in a graphic way that would facilitate their understanding for readers.
Response: It has been clearly mentioned in the Figure.3 and Figure.4.
Round 2
Reviewer 1 Report
The authors have addressed most of the comments and suggestions. The quality of the article has significantly improved.
However, several aspects still need to be considered:
- First, the quality of Figure 4 has not been improved in the revised version.
- The strengths and limitations of the proposed method in comparison with previous approaches has not been sufficiently addressed.
Author Response
The authors have addressed most of the comments and suggestions. The quality of the article has significantly improved.
However, several aspects still need to be considered:
- First, the quality of Figure 4 has not been improved in the revised version.
- The strengths and limitations of the proposed method in comparison with previous approaches has not been sufficiently addressed.
Response: The figures and the appropriate strength and limitations are added in the appropriate places.
Reviewer 2 Report
The authors have put a lot of effort into editing the paper.
But your paper needs a lot of corrections about the figures, and some figures was not reflect the reviewer's comments.
1. The figures in the paper are difficult to understand and very abstract. The Figures should be more specific. And there must be connectivity between the modules in the each figure. Try to improve the quality of the figures. (Figure 1~4)
2. In particular, in figure 4, there are only modules listed, and there is no connection and explanation between modules.
Author Response
The authors have put a lot of effort into editing the paper.
But your paper needs a lot of corrections about the figures, and some figures was not reflect the reviewer's comments.
- The figures in the paper are difficult to understand and very abstract. The Figures should be more specific. And there must be connectivity between the modules in the each figure. Try to improve the quality of the figures. (Figure 1~4)
- In particular, in figure 4, there are only modules listed, and there is no connection and explanation between modules.
Response: As per your comment, the figures are corrected with better clarity
Reviewer 3 Report
- Please clearly write your contribution from the beginning and the conclusion. For example, your proposed foumulars, functions and simulation method etc. This journal emphasizes very clear simulation methods for many scientists and engineers etc.
Author Response
- Please clearly write your contribution from the beginning and the conclusion. For example, your proposed foumulars, functions and simulation method etc. This journal emphasizes very clear simulation methods for many scientists and engineers etc.
Response: The contribution has been added with better clarity to make the concept clear to the readers.
Reviewer 4 Report
The reviewed paper increased its readability and value after corrections were made by the authors. Unfortunately, the authors have not completely corrected all the shortcomings.
Text formatting (including punctuation) still needs improvement, and the quality of the figures is questionable (Fig 1 and Fig. 4). In addition, the variables in the text should be written in italics.
Despite the response to one of the comments regarding the description of the formulas, there is still no explicit designation, e.g. d(CDj,Best). So, does d(CDj,Best) = CDj,Best?
The paper lacks a wider reference to other algorithms the reasons for choosing specific ones by the authors. It seems to me that it would be worth referring to other groups of algorithms and models, and justifying the Inspired Bio-computing Model more precisely.
Moreover, the summary should also include a reference to the simulation results obtained.
Thus, the paper still needs improvement before it can be published.
Author Response
The reviewed paper increased its readability and value after corrections were made by the authors. Unfortunately, the authors have not completely corrected all the shortcomings.
Text formatting (including punctuation) still needs improvement, and the quality of the figures is questionable (Fig 1 and Fig. 4). In addition, the variables in the text should be written in italics.
Despite the response to one of the comments regarding the description of the formulas, there is still no explicit designation, e.g. d(CDj,Best). So, does d(CDj,Best) = CDj,Best?
The paper lacks a wider reference to other algorithms the reasons for choosing specific ones by the authors. It seems to me that it would be worth referring to other groups of algorithms and models, and justifying the Inspired Bio-computing Model more precisely.
Response: As per the instructions, all the figures have been drawn with clear view. And the corresponding smaller changes have also been done.
Moreover, the summary should also include a reference to the simulation results obtained.
Response: The summary and the comparisons have been added in the proper places according to the sections.
Thus, the paper still needs improvement before it can be published.
Round 3
Reviewer 2 Report
The authors have put a lot of effort into writing a paper, but there is still much to be done.
I would like to suggest that the authors studies more about how to draw and the overall composition of the paper, especially for Figures and submit it again as a better paper.
